# EASING NON-CONVEX OPTIMIZATION WITH NEURAL NETWORKS

**David Lopez-Paz**[1*], **Levent Sagun**[2*]
[1]Facebook AI Research, [2]Ínstitut de Physique Théorique, CEA, CNRS
dlp@fb.com, levent.sagun@ipht.fr

## ABSTRACT

Despite being non-convex, deep neural networks are surprisingly amenable to optimization by gradient descent. In this note, we use a deep neural network with $D$ parameters to parametrize the input space of a generic $d$-dimensional non-convex optimization problem. Our experiments show that minimizing the over-parametrized $D \gg d$ variables provided by the deep neural network eases and accelerates the optimization of various non-convex test functions.

## 1 INTRODUCTION

We consider the ubiquitous problem of optimization

$$x^\star = \arg \min_x f(x), \tag{1}$$

where $f : \mathcal{X} \to \mathcal{Y}$ is a non-convex differentiable function. Without losing much generality, we consider the case where $\mathcal{X} = [0, 1]^d$, and $\mathcal{Y} = [0, 1]$.

**Contribution** We demonstrate that it is possible to use deep neural networks (Goodfellow et al., 2016) to over-parametrize the $d$-dimensional input space of non-convex optimization problems (1). Such over-parametrization eases and accelerates the optimization process. Our method is inspired by the fact that deep neural networks, although non-convex, are surprisingly well-behaved when optimized by gradient descent. One possible reason behind this phenomenon is that the parameters of deep neural networks are highly redundant (Denil et al., 2013). Such a redundancy through over-parametrization is one of the reasons for the ease of training in deep neural networks (Livni et al., 2014). Intuitively, over-parametrization is opportunity: damage coming from some poorly initialized or optimized parameters can be undone by the proper optimization of others. In this note, we question whether we can make use of this insight in generic non-convex optimization problems.

**Motivational Example** Consider the 1-dimensional parabola $f(x) = x^2$ over $[-1, 1]$. We can reparametrize the domain of $f$ by a 1-hidden layer network with two parameters, one fixed input, and linear activations: $x = w_1 w_2 \tilde{x}$. Then, $f(x) = f(w_1 w_2 \tilde{x})$ can be seen as a function of $(w_1, w_2) \in \mathbb{R}^2$. Figure 1 shows how the over-parametrization of $f$ from 1-dimension to 2-dimensions introduces a continuum of equivalent solutions, steeper descent directions, and a flatter basin of global minima.

## 2 METHOD

We translate the optimization problem (1) into

$$z^\star, \theta^\star = \arg \min_{z, \theta} \frac{1}{n} \sum_{i=1}^{n} f(g_\theta(z_i)), \tag{2}$$

where $g : \mathbb{R}^k \to [0, 1]^d$ is a deep neural network with parameters $\theta \in \mathbb{R}^p$, and accepting inputs $z = (z_1^\star, \ldots, z_n^\star) \in \mathbb{R}^{n \times k}$. After approximating the non-convex optimization problem (2), we

---

*Alphabetical, joint first authors.

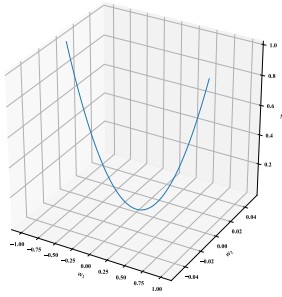
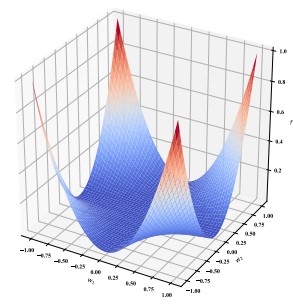

(a) Parabola, original 1d domain        (b) Parabola, overparametrized 2d domain

Figure 1: A very simple visual example of what over-parametrization does to the loss-surface: All of the points on the lines $w_1 = 0$ and $w_2 = 0$ correspond to the solution of the original problem.

return the solution: $x^\star = \min(g_{\theta^\star}(z_1^\star), \ldots, g_{\theta^\star}(z_n^\star))$ as our best estimate of the location of the global minima of the objective function $f$.

In a nutshell, we have translated the $d$-dimensional non-convex optimization problem (1) into the equivalent non-convex $D$-dimensional non-convex optimization problem (2), where $D = nk + p \gg d$ is the number of neural network parameters $\theta$ plus the number of variables contained in the $n$ neural network $k$-dimensional inputs $z$. Some observations are in order:

- We assume that the original function has non-degenerate critical points (as in the case of our examples in this note).
- The set of points in the weight space corresponding to *different* minima of the original function form an equivalence relation.
- Two points taken from different equivalence classes in the weight space has a barrier between them. In general, different solutions are not necessarily connected (in contrast to what's claimed for deep neural networks with convex loss criteria (Freeman & Bruna, 2016; Sagun et al., 2017)).
- The minimum height of the barrier between such weights is the same as the minimum height of the barrier between the corresponding minima of the original function.

The problem (1) runs $n$ optimization problems in parallel, one per input in the "minibatch" $z$. These $n$ optimization problems run efficiently by virtue of vectorized (as well as GPU) implementations of deep learning software (Paszke et al., 2017). As it is the case with (1), the problem (2) can be optimized using any gradient-based optimizer. Finally, if the $d$ variables of $f$ are related to each other, the network $g$ may capture these dependencies, accelerating the optimization process.

## 3   EXPERIMENTS

Due to simplicity and space limitations, we restrict ourselves to vanilla gradient descent. For the deep optimizer, this leads to the updates

$$\theta^{t+1} \leftarrow \theta^t - \alpha_t \cdot \frac{\partial \left\{ \frac{1}{n} \sum_{i=1}^n f(g_{\theta^t}(z^t)) \right\}}{\partial \theta^t}, \quad z^{t+1} \leftarrow z^t - \alpha_t \cdot \frac{\partial \left\{ \frac{1}{n} \sum_{i=1}^n f(g_{\theta^t}(z^t)) \right\}}{\partial z^t},$$
$$x^{t+1} \leftarrow \min(g_{\theta^t}(z_1^t), \ldots, g_{\theta^t}(z_n^t)),$$

where $z^0, \theta^0$ are drawn from a Gaussian distribution, as implemented in PyTorch 0.4.0 (Paszke et al., 2017). The best initial learning rate $\alpha^0$ is chosen from $\{10^0, 10^{-1}, \ldots, 10^{-10}\}$, and multiplied by a factor 0.9 after each iteration without improvement. Our preliminary experiments showed that this simple strategy outperforms a variety of complex state-of-the-art optimizers (Wilson et al., 2017).

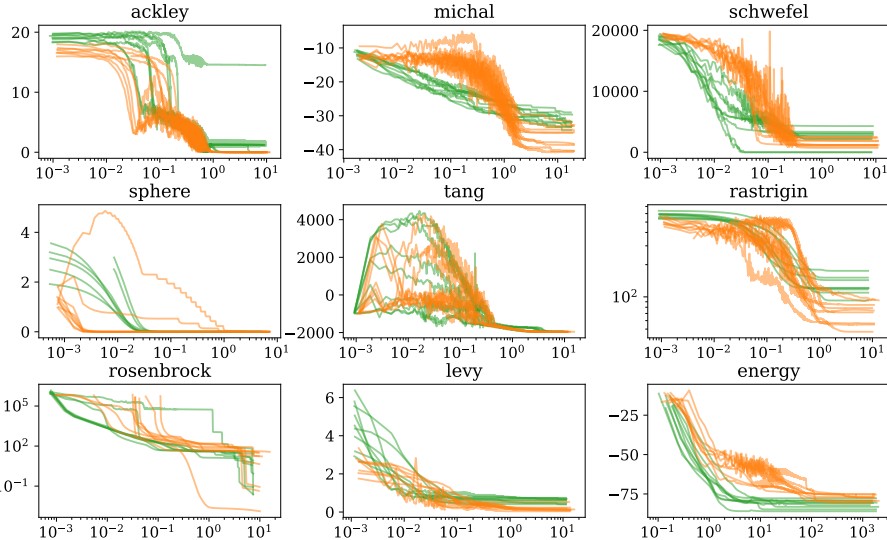

Figure 2: Wall log-time versus objective value, for shallow (green) and deep (orange) optimizers.

## 3.1 TEST FUNCTIONS

We perform non-convex optimization of the Ackley, Levy, Rastrigin, Schwefel, Sphere, Rosenbrock, Styblinski-Tang, and Michalewicz functions (Surjanovic & Bingham, 2013). Also, we perform non-convex optimization of the Hamiltonian of a 3-spin spherical glass model (Auffinger et al., 2013). We scale the suggested function input supports (Surjanovic & Bingham, 2013) to $[0, 1]^d$, and scale their output to map the recommended support into $[0, 1]$.

## 3.2 NEURAL NETWORK ARCHITECTURE

The deep optimizer $g_\theta(z)$ is a feed-forward neural network composed of the following modules:

$$\text{EmbeddingLayer}(n, n_h) \rightarrow \Big[ \text{Linear}(k, n_h) \rightarrow \text{ReLU}() \Big] \times n_l \rightarrow \text{Linear}(n_h, d).$$

The input to the previous architecture are $n$ $n_h$-dimensional embeddings, that is, $n$ $n_h$-dimensional *trainable* inputs. These inputs are fed to $n_l$ hidden layers of $n_h$ ReLU neurons, followed by one final linear transformation into the target $d$-dimensional space. The ten runs reported in Figure 3.3 are for $n = 50$, $n_h = 10$, $d = 50$, and $n_l = 1$. Our source-code is available at `https://github.com/facebookresearch/DeepConvexity`.

## 3.3 RESULTS

After initializing all optimizers to produce the same initial $x^0$ value, we run the shallow optimizer $10,000$ iterations for the test functions, and $1,000$ iterations for the Hamiltonian. We then run the deep optimizer for the same clock time, and repeat this process for 10 different random initializations for the test functions (or three for the Hamiltonian). Figure 3.3 shows the evolution of the *minimum* error over all the $n$ embeddings for the Rastrigin function and the Hamiltonian versus the logarithm of number of iterations. Since *there is no stochasticity* in these experiments, the minimum value in each plot (from all curves at all locations) would be reported as the best estimate of the global minima of the non-convex optimization problem.

**Conclusion** The deep optimizer usually outperforms vanilla gradient descent in terms of best minima found. Perhaps surprisingly, the deep optimizer is also *much faster* as per clock time, as in most cases, the deep optimizer converges 10 times faster. Although more extensive experiments are necessary, it seems that deep neural networks provide a powerful reparametrization that allows to ease and accelerate the optimization of non-convex problems.

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
