# OpenReview forum: "Easing non-convex optimization with neural networks"
_ICLR.cc/2018/Workshop — Accept_

### Official Review · AnonReviewer1 · 2018-02-28
**Interesting Results on Overparametrization**

**Rating:** 6
**Confidence:** 4

**Review:**

This paper provides interesting new angle on over-paramterization. Existing works mostly try to explain why over-parametrization works for neural network. This paper tries over-paramterization on classical optimization problems and demonstrate its effectiveness.
The experiments are interesting. However, I hope to see more theoretical (not necessary rigorous) analysis.

---

### Official Review · AnonReviewer3 · 2018-03-09
**Overall good but needs some improvements**

**Rating:** 7
**Confidence:** 4

**Review:**

This paper considers the problem of solving non-convex problems using deep neural network. The authors propose that the deep neural networks can ease and accelerate the optimization process because of the redundancy of parameters. This idea is interesting and the results are promising. The experiments show that if D>>d, using deep neural networks can achieve faster convergence. My major concerns are as follows.
(1) The test data is small. The authors may want to conduct experiments on large scale real world data.
(2) The authors may try their method to solve more non-convex problems.

---

### Official Review · AnonReviewer2 · 2018-03-10
**Review for "Easing non-convex optimization with neural networks"**

**Rating:** 6
**Confidence:** 4

**Review:**

This paper studies using neural networks to assist non-convex optimization. The main idea is to replace the parameter vector x (to be optimized) by the output of a neural network g(z), then optimize both z and the parameters of network g. The paper also proposes a parallelized scheme. That is, instead of optimizing f(g(z)), it defines the objective function to be f(g(z_1)) + ... + f(f(z_n)), then jointly optimize (z_1,...,z_n) and the parameters of g. Then the best solution among g(z_1),...,g(z_n) is taken as the final solution.

The idea behind the optimization algorithm is interesting. The problem will be solved if one of the n sub-problems for optimizing f(g(z_1)), ..., f(g(z_n)) is solved. The computation for solving these problems are correlated. Indeed, if one of z_1, ... , z_n reaches a region of low function value, it will help shaping the shared network g, which can help other sub-problems attaining a low function value as well. Another benefit of this optimization scheme is to utilize the existing linear algebra and parallel computing frameworks, which have been highly optimized for neural network computation. Since the algorithm is very simple, it is widely applicable to a broad range of non-convex optimization problems.

Although the idea is promising, the empirical justification is preliminary. There is no experiments on real problems, or large-scale synthetic data.

In addition, the proposed method should be compared with the following simple baseline:

optimize f(x_1) + .... + f(x_n)

where x_1,...,x_n have different random initializations. The hyper-parameter n should be set such that the total number of parameters in the above problem is equal to the total number of parameters in the proposed neural network method. The proposed method should be compared with this baseline in CPU time.

---

### Decision · Program_Chairs · 2018-03-20
**ICLR 2018 Workshop Acceptance Decision**

**Decision:**

Accept

**Comment:**

Congratulations, your paper was accepted to the ICLR workshop.